# Determinants of Dividend Policy: The Case of the Casablanca Stock Exchange

Reda Louziri * and Khadija Oubal

Faculty of Legal, Economic and Social Sciences—Souissi at Mohammed V University in Rabat, Rabat 8007, Morocco
* Correspondence: redalouziri@gmail.com; Tel.: +212-650-21-56-82

**Abstract:** This article investigates the determinants of dividend policy on the Casablanca stock exchange. The variables tested were based on the main theories of dividend policy, and the fixed effect model was used to test panel data over a period of 16 years from 2003 to 2018. The eight independent variables tested were profitability, firm size, retained earnings, firm age, leverage, growth opportunities, price to earnings (P/E) and a dummy variable introduced for financial companies. To corroborate the results, two proxies were used to test the dependent variable: dividend yield and payout ratio. The results led to the identification of three significant determinants of dividend policy, which are firm age, growth opportunities and firm size. The negative correlation between the variables of firm size and firm age with dividend policy is explained by signaling theory. On the other hand, the negative correlation between growth opportunities and dividend payments is predicted by different theories, such as agency theory, financial flexibility theory and life cycle theory. This study provides insights for investors, analysts and researchers into dividend policy determinants on the Casablanca stock exchange based on firms' characteristic variables.

**Keywords:** dividend policy determinants; dividend yield; payout ratio; firm age; growth opportunities; firm size

## 1. Introduction

The main objective of this article was to identify the determinants of dividend policy in firms listed on the Casablanca stock exchange, which is of major interest to investors, shareholders, managers and financial analysts. This financial decision continues to attract more academic researchers, given the lack of consensus on the factors explaining dividend payments. In fact, dividend policies adopted by listed companies continue to be heterogeneous and differentiated, despite the numerous academic and empirical studies conducted on this subject. Indeed, several researchers have tried, for several decades, to present and support their theories about dividend policy. Different determinants were then presented in order to explain dividend payments; however, none of the theories of dividend policy alone provide a global explanation of dividend payment choices. Thus, the academic debate on dividends continues to draw new researchers, who aim to provide new answers to this complex financial decision in order to elucidate why companies pay, pay more or do not pay dividends.

During the period from 2003 to 2018, the average dividend yield ratio reached 8.16% (929 observations) and the average payout ratio was equal to 1.36 (921 observations). However, the standard deviations of these two ratios were high and exceeded 17.36% for the dividend yield and 2.91 for the payout ratio. This situation demonstrates that dividend payments differ widely between listed companies that pursue heterogeneous dividend policies.

The Casablanca stock exchange presents some specific characteristics. Indeed, the number of listed firms is very limited and reaches 70 firms on average compared to other

African stock markets, such as in Egypt or South Africa, where the number of firms exceeds 200. Moreover, the Moroccan stock market includes a large number of financial firms. In our sample, the number of these firms was 18 (banks, insurance companies and credit institutions). In addition to this, an empirical study of the impact of firms' characteristics on dividend payments, which tests the different theories of dividend policy, has not been carried out yet. Therefore, this study would provide insights for investors, analysts and researchers into the determinants of dividend policy on the Moroccan stock market based on firms' characteristics, which is of major interest to define portfolio policies chosen by investors and to provide a better understanding for analysts of dividend policy in Morocco. Moreover, this study would provide a basis for academic and empirical debate about dividend policy in Morocco, and the significant variables could be used as control variables to conduct further empirical studies about the effect of corporate governance (ownership and board of directors' structure) and CEO characteristics on dividend payments in the Casablanca stock exchange.

The academic debate on dividend policy continues to experience consistent evolution, given the mobilization of new theories that continue to offer new answers to the choices of dividend payments. Two theories, the "bird in hand" by Lintner (1956) and irrelevance theory by Miller and Modigliani (1961), launched this extensive debate. The theory of "bird in hand" suggested that investors prefer firms that pay dividends, due to the certainty of dividend payments, while Miller and Modigliani (1961) indicated that the payment of dividends was not necessarily relevant, since it would not impact the value of the share. The residual theory of dividends (Partington 1985; Baker 2009) stated that dividends were made after financing all profitable investments. Another explanation was provided by signaling theory (Bhattacharya 1979; John and Williams 1985) based on information asymmetry problems. Therefore, managers would pay dividends in order to send a signal to investors about their ability to generate future profits. Agency theory (Jensen and Meckling 1976; Jensen 1986) then proposed a different alternative by introducing agency costs that could arise due to conflicts of interest between shareholders and managers. The reduction in the available cash flow through dividend payments would limit agency costs and avoid unprofitable investments. As for the theory of financial flexibility (Rozeff 1982), it focused on transaction costs to justify dividend distribution. Indeed, internal funds, considered to be less expensive than external funds, would finance growth opportunities and, therefore, dividend payments would be limited. Life cycle theory (Grullon et al. 2003; DeAngelo et al. 2006) proposed an analysis of the development stages of the firm to elucidate dividend policy. Thereby, companies in a mature stage present low levels of growth opportunities and would pay higher dividends.

In order to conduct an empirical study on dividend policy determinants on the Casablanca stock exchange, this article will first present a literature review of the theories of dividend policy. Then, the second section will present the model, the variables, the sample and the hypotheses of the predicted correlation between each independent variable and the dependent variable based on the theoretical framework. The last section will present the empirical results by identifying the determinants of dividend policy in the Casablanca stock exchange.

This study enabled us to identify three significant determinants of dividend policy in the Casablanca stock exchange, which are firm age, growth opportunities and firm size. While none of the theories presented in this article explain all the results obtained, the complementarity of different theories enables us to justify our empirical results.

## 2. Literature Review of Dividend Policy

Pyles (2013) defines dividends as payments made to shareholders based on the net profits made by the company. When a company makes profits, it has two major choices in distributing them to shareholders: (1) to keep the profits made within the company, which will increase the value of the share; (2) to distribute the profits through dividend payments or repurchase its own shares.

The theories of the "bird in hand" by Lintner (1956), the irrelevance theory by Miller and Modigliani (1961), and the residual theory by Partington (1985) launched the debate about dividend policy. Nevertheless, several theories attempted to provide further explanation to understand why firms pay or do not pay dividends, such as agency theory, signaling theory, financial flexibility theory and life cycle theory. Therefore, this section will present a literature review on dividend policy by providing the main assumptions of these theories.

*2.1. Agency and Transaction Costs, Information Asymmetry and Life Cycle Theory as New Alternatives to Explain Dividend Policy*

2.1.1. Signaling Theory

Bhattacharya (1979) and John and Williams (1985) indicated that managers, as holders of private information, choose the amount of dividends in order to send a signal to investors about the potential future profits of the firm. This signal aims to reduce the degree of information asymmetry between managers and investors. Managers would also be more likely to use this mechanism when they believe that the value of the share is undervalued. Bhattacharya (1979) pointed out that, despite the tax disadvantage of dividend payments, companies that predict satisfactory levels of future cash flows tend to distribute an increasing level of dividends in order to signal to the market that they possess a strong future earnings capacity and that they could maintain the same levels of dividend payments.

Lang and Litzenberger (1989) analyzed the reaction of the market following changes in dividends by using Tobin's Q variable (market value of equity/book value of equity). They found that the market reacts positively and mainly to dividend distributions made by companies whose growth opportunities are weak. Furthermore, they indicated that this market reaction is evidence of signaling theory.

Nevertheless, Fama and French (2001) suggested that the main problem of this theory is its inability to explain why older, larger and more profitable companies pay more dividends, as these are supposed to experience fewer information asymmetry problems and should pay less dividends than the least profitable and small firms.

2.1.2. Agency Costs and Dividend Policy

Jensen and Meckling (1976) assumed that dividend payments are made in order to reduce agency costs. Shareholders would prefer the payment of dividends so as to limit the available cash flow that could be invested in unprofitable projects, due to the conflict of interest between managers and shareholders. In addition to this, Jensen (1986) presented the free cash flow hypothesis, which indicates that the free cash flow should be distributed in full to the shareholders. Thus, the available free cash flow, after financing profitable investments and paying debt engagements, should not be maintained in the firm, so as to avoid decisions made by managers that may lead to a reduction in the value of the firm (Easterbrook 1984; Jensen 1986).

Easterbrook (1984) noted that dividends could make it possible to align the interests of managers with those of investors, thus making it possible to reduce agency costs. Indeed, two explanations have been presented by the author to explain why dividends should be paid. The first is related to the cost of control that is reduced by dividend payments. In order to control the managers, shareholders would reduce the available cash flow to avoid unprofitable investments made by the managers. The second argument is about the cost of managers' risk aversion. Unlike investors, who generally have diversified portfolios, the wealth of managers can be mainly linked to a single company. Consequently, the distribution of an optimal level of dividends should make it possible to control these two costs with the objective of limiting internal funds. Moreover, the author highlighted the fact that dividends should be distributed only to reduce agency costs, because they do not create additional value for shareholders.

### 2.1.3. Financial Flexibility and Transaction Costs

High dividend payments may reduce agency costs, but they could increase external financing costs (Rozeff 1982). The "pecking order" theory of Myers and Majluf (1984) stated that managers would prefer the use of internal cash flow to other means of financing, such as debt or equity, which have higher costs.

Companies that display or plan to achieve high growth rates are supposed to maintain low levels of dividend distribution in order to be able to finance their investments through internal funds, which are less expensive. As such, Holder et al. (1998) found that riskier companies (that have higher external financing costs) with high levels of growth distribute fewer dividends. They also indicated that large companies, which are more mature, have easier access to financial markets and are, therefore, less dependent on their internal resources. Furthermore, companies with more tangible assets can also access external financing more easily and could, therefore, pay more dividends.

Some studies used the P/E proxy for the level of risk of the company. According to Myers and Bacon (2004), a high P/E would indicate that the company is less risky. Therefore, these companies can afford to pay more dividends due to their less costly access to external funds. As such, their results demonstrated a positive correlation between P/E and dividends. However, the study by Franklin and Muthusamy (2010) found a negative correlation between this P/E ratio and the payout ratio.

### 2.1.4. Life Cycle Theory

Life cycle theory (Grullon et al. 2003; DeAngelo et al. 2006) assumes that mature companies, which present fewer growth opportunities, pay high dividends. On the other hand, companies in development stages would pay fewer dividends in order to finance their growth opportunities.

Fama and French (2001) explained the decline of American companies' dividends by the increase in the number of small firms that present low levels of profitability but high growth rates. These results suggest that companies in growth stages are less likely to pay dividends.

Furthermore, DeAngelo et al. (2006) argued that mature companies are more likely to pay high dividends, since the decision to pay them is linked to the amount of retained earnings. Therefore, firms start to pay dividends when they show a decline in their growth rate and profitability forecasts (Fama and French 2001; DeAngelo et al. 2006).

### 2.2. Empirical Results of Previous Studies on the Determinants of Dividend Policy

Different studies carried out on many financial markets have tried, over several years, to identify the determinants of dividend policy. As such, Holder et al. (1998) tested several variables of 477 companies. They found that the size variable presents a significant positive correlation with the payout ratio, which means that larger companies pay more dividends compared to smaller ones. This result is consistent with life cycle theory, which assumes that mature companies pay more dividends because their growth opportunities are decreasing. This result is consistent with other research, which has also confirmed that larger companies pay more dividends (Vogt 1994). In addition to this, riskier companies pay low levels of dividends and companies that show high growth sales rates also show low payout levels.

Grullon et al. (2003) tested a sample of companies from 1963 to 1997 in the American financial market, and they specified that dividend payments do not provide a signal on the capacity to generate future profits and achieve better performance. Their results showed a negative correlation between dividend payments and future profitability. These results are corroborated by the study of Benartzi et al. (1997), who focused on the American financial market and concluded that the payment of dividends did not offer signaling information about the future returns of the firm and its ability to generate profits.

Aivazian et al. (2003) studied the determinants of dividend policy in eight emerging or developing countries in order to compare the results obtained with those from studies carried out in developed countries. The variables tested were the risk of the activity, volatility of profits, size, tangibility of assets, financial profitability (return on equity), price to book ratio and leverage. Some results were similar to those generally found in the financial markets of developed countries, such as the positive relationship between dividends and profitability (other studies in developing countries corroborate these results: Naceur et al. (2006) for Tunisian companies and Sun et al. (2010) for Chinese ones). They also found a negative relationship between earnings volatility and dividend distribution, which suggests that companies with unstable earnings prefer not to pay dividends due to the negative effect that could arise when future decreases in dividends might be needed. They also considered the problem of access to financial markets by assuming that companies that have more tangible assets could more easily access external funds. However, the correlation between assets' tangibility and dividends is not significant. They also found a negative relationship between dividends and leverage. Companies with higher leverage tend to pay low amounts of dividends. Managers, considered risk-averse, prefer to retain the cash flow internally in order to face any financial distress risks and pay debts. Moreover, the growth opportunities (P/B) variable showed a positive correlation with dividends, which would support signaling theory.

Ho (2003) studied more than 2000 observations in the Australian and Japanese financial markets and found a positive correlation between company size and dividend payments. Moreover, the study found a positive correlation between dividends and the two variables of liquidity and risk in the Japanese financial market. Thus, riskier and more liquid firms tend to distribute higher dividends. In addition, the results showed that the industry has a significant impact on dividend policy in the two financial markets.

Myers and Bacon (2004) examined more than 480 companies in the American financial market for a period of five years. They tested several variables, such as the P/E ratio, profitability, leverage and expected growth in earnings. The study showed a positive relationship between P/E and dividends, explained by the risk level of the company. In addition to this, Krishnan and Chen (2017) found a positive link between dividends and P/E.

Furthermore, Baker and Smith (2006) used leverage in their model and found that leveraged firms pay low amounts of dividends. This study confirmed the use of dividends as a substitute for debt to control corporate managers.

The study of Al-Malkawi (2007) investigated the Jordanian financial market during the period from 1989 to 2000. He identified three significant variables, which are firm size, profitability and firm age. According to the author, these results support the agency costs hypothesis.

Denis and Osobov (2008) analyzed dividend policy on an international scale, using cross-sectional and time-series methods, over the period from 1989 to 2002 in six countries (Japan, Germany, United States, United Kingdom, France and Canada). Their results indicate that larger and more profitable companies pay more dividends. These results confirm the theories of agency costs (Jensen 1986) and life cycle (DeAngelo et al. 2006), but they are in contradiction with signaling theory, which supposes that companies suffering from information asymmetry problems (small and less profitable companies) send a signal to the market about their future profits through higher dividends. Moreover, the probability of paying dividends is strongly related to the ratio of retained earnings to equity. For growth opportunities, the study found that dividend payers in the UK, US and Canada had low levels of growth opportunities, while those in Japan, Germany and France presented higher levels in terms of growth opportunities. Furthermore, the study indicated that dividends did not decline throughout this period, because they were mainly concentrated in the largest and most profitable companies.

The study of Chay and Suh (2009) supports life cycle theory, claiming that equity earned over the lifetime of the company positively impacts the amount of dividends paid.

Moreover, the level of cash flow volatility negatively impacts the amount of dividends, since managers can expect a significant drop in cash flow and, therefore, would have to finance dividends through external funds in order to maintain the same level of payment. However, the study did not find a significant relationship between growth opportunities and dividends.

In the Moroccan stock market, two previous studies (Aguenaou et al. 2013; Mossadak et al. 2016) tried to identify dividend policy determinants; however, they focused on the ownership effect on dividend policy and used a few firms' characteristics only as control variables. Indeed, Aguenaou et al. (2013) used size, leverage and earnings per share as control variables and did not find a significant correlation between these variables and dividends. The research of Mossadak et al. (2016) used size, return of equity and leverage as control variables. Their results indicate that leverage has a significant negative correlation with dividend payments, while the two other firm characteristic variables do not show a significant relationship with dividend policy.

Al-Sabah (2015) conducted a study on the financial market in Kuwait over a period of 5 years. The results showed a negative correlation between leverage and dividends and a negative correlation between firm age and dividend payments. However, Mili et al. (2017) revealed a positive correlation between dividends and firm age. The researchers explained this relationship through the agency costs hypothesis, since most mature companies suffer more from agency cost problems and pay high dividends to control the managers. This result is also supported by life cycle theory, which predicts that mature firms (higher age) pay more dividends.

Al-Najjar and Kilincarslan (2018) conducted a study on the stock market of Istanbul during the period from 2003 to 2012. The results indicate that more profitable, larger and mature companies are more likely to pay dividends and distribute higher amounts. On the other hand, leveraged firms with greater growth opportunities are less likely to pay high dividends.

Grace et al. (2019) looked at the dividend policy of 36 industrial companies over a 20-year period (from 1997 to 2006). The results show that leverage and sales growth present positive correlations with the payout ratio. However, the operating cash flow, size and earnings per share variables present a negative correlation with dividends.

Danila et al. (2020) conducted a study of listed companies in the Indonesian financial market during the period 2007–2017. The results reveal a negative correlation between growth opportunities and dividend yield. Moreover, profitability showed a positive correlation with dividend policy.

Al-Sawalqa (2021) investigated the impact of retained earnings, share book value, asset turnover, firm age and size on dividend policy. The results for 179 non-financial companies over a period of five years show that share book value and retained earnings present a positive correlation with dividends, while the other variables show no significant results. The author specifies that these results support life cycle theory.

The study of Setiawan and Vivien (2021), focusing on dividend policy determinants in Indonesia, was conducted over a 4-year period. They found that profitability and firm size are significant determinants and they both show a positive correlation with dividend policy.

Kengatharan (2021) studied dividend policy determinants in Sri Lanka during the period 2013 to 2017. His research revealed that profitability, risk and earnings per share present a negative correlation with dividend policy.

Kahraman (2021) studied dividend policy in the Indian financial market during the period from 2000 to 2021. The results indicate that profitability, leverage and liquidity are major factors that impact dividend policy decisions.

Ali (2022) studied dividend policy in the period of COVID-19 and indicated that dividend payments decreased due to this health crisis among the G-12 countries. The study tested the following variables: leverage, profitability, assets turnover, earnings prospects, size and market to book ratio. The significant variables revealed by this study were as follows: profitability, size, leverage and earning prospects.

*2.3. Impact of Corporate Governance and CEO Characteristics on Dividend Policy*

Because our study is the first one that focuses on dividend policy determinants in the Casablanca stock exchange (to our knowledge), we attempted first to test the impact of the firm's characteristics on dividend policy. Nevertheless, it should be noted that other studies have attempted to test the impact of other variables on dividend policy, such as corporate governance variables and CEO characteristics. Indeed, based on agency theory, La Porta et al. (2000) and Sawicki (2009) have presented the outcome model, suggesting that well-governed firms would pay higher dividends in order to confirm to the shareholders that they are taking decisions in their best interests. On the other hand, La Porta et al. (2000) and John and Knyazeva (2006) suggested that firms, based on the substitute model, would pay more dividends in order to reduce the cash flow available for the managers internally in order to limit agency costs. Different authors have then investigated the correlation between corporate governance variables, such as Mili et al. (2017), who found a positive correlation between institutional investors and payout ratios, or the study of Bataineh (2020), who showed a negative correlation between dividends and the presence of foreign ownership.

In addition to these, some other studies included the CEO's characteristics in order to understand dividend policy. The research of Al-Ghazali (2014) tested the following variables and their effect on dividend policy: overconfident CEOs, the CEO's age, the CEO as the founder of the firm and the CEO's nationality. Other authors have also tried to test the relationship between CEO characteristics and dividends, such as McGuinness et al. (2015), who found a positive correlation between the CEO's age and dividend payments. However, studies of the effect of the CEO's characteristics and dividend policy remain rare compared to those that test the firm's characteristics or the corporate governance impact of dividend policy.

## 3. Variables, Sample and Regression Model

After presenting the different theories of dividend policy, this section will focus on the choice of the sample, the variables tested and the regression model. It will then detail the predicted correlation between each independent variable and the dependent variables.

*3.1. Variables*

3.1.1. Dividend Distribution

One of the most common measures is the dividend yield ratio, which is calculated as follows: dividend yield = dividend per share/share price. The ratio takes into consideration the share price, which means that it is subject to external financial market factors that can influence dividend policy, but it could present more information about the company. Indeed, several studies have used this ratio, such as those by Miller and Scholes (1982), Odak (2006), Al-Najjar and Hussainey (2009), Mili et al. (2017), Subramaniam (2018) and Bataineh (2020).

A second proxy will then be used to test the robustness of the statistical tests. The payout ratio measures the amount of dividends paid to shareholders compared to the generated profit (Damodaran 2010). It is calculated as follows: payout ratio = dividend per share/earnings per share. This ratio can be independent of external factors (Penman 2010), given the use of book data. Several studies chose this ratio, such as those of Rozeff (1982), Mitton (2004), Baker and Smith (2006), Lam et al. (2012), Le and Le (2017), Chen et al. (2017) and Sari (2018).

3.1.2. Profitability

We chose the return on equity (ROE) proxy to test the relationship between profitability and dividend policy. This ratio calculates the profitability of the company based on the equity invested (Al-Kuwari 2009). Various studies used this ratio: those of Nissim and Ziv (2001), Aivazian et al. (2003), Kania and Bacon (2005), Abdelsalam et al. (2008) and Adil et al. (2011).

Several studies showed a positive relationship between the two variables, such as those of Nissim and Ziv (2001), Aivazian et al. (2003), Denis and Osobov (2008), Al-Kuwari (2009), Rehman (2012), Issa (2015), Al-Najjar and Kilincarslan (2018), Eluyela et al. (2019), Grace et al. (2019) and Danila et al. (2020). However, other researchers found a negative relationship between profitability and dividends, such as Grullon et al. (2003), as well as Kania and Bacon (2005). Moreover, some authors did not find a correlation between dividend distribution and profitability (Ho 2003; Adil et al. 2011).

### 3.1.3. Firm Size

We used the logarithm of total assets as a proxy of size, which is one of the most common measures in empirical research (Dang et al. 2018). Indeed, other proxies could be used for firm size, such as total market capitalization (Al-Kuwari 2009; Dang et al. 2018). However, this proxy is influenced by different market effects and can overestimate or underestimate based on financial economic cycles. Moreover, this proxy provides information about growth opportunities (Dang et al. 2018), which is already captured by another independent variable (P/B ratio). The third commonly used proxy is total sales (Dang et al. 2018), which measures product market competition, but it is not forward-looking. Moreover, for financial firms, which are very common on the Casablanca stock exchange (18 firms), this proxy could not provide enough information about their size because of their industry.

Other, less commonly used proxies could also be chosen, such as total employees. However, this proxy does not include part-time employees (Hart and Oulton 1996).

Indeed, multiple studies that investigated dividend policy used total assets as a proxy of size, namely those of Holder et al. (1998), Koch and Shenoy (1999), Denis and Osobov (2008), Al-Najjar and Hussainey (2009), Al-Shubiri et al. (2012), Subramaniam (2018) and Al-Sawalqa (2021). Moreover, to reduce the effect of high amounts of total assets, we have introduced the logarithm of these assets, as described by Koch and Shenoy (1999), Chay and Suh (2009), Adil et al. (2011), Al-Shubiri et al. (2012) or Subramaniam (2018).

Several studies showed a positive relationship between size and dividends, including those of Holder et al. (1998), Ho (2003), Al-Malkawi (2007), Denis and Osobov (2008), Al-Kuwari (2009), Sun et al. (2010), Eriotis (2011), Lumapow and Tumiwa (2017), Al-Najjar and Kilincarslan (2018) and Grace et al. (2019). These results support agency theory and life cycle theory. Nevertheless, Javid and Ahmed (2009), Al-Shubiri et al. (2012) and Nazir et al. (2012) found a negative relationship between dividends and size, which supports signaling theory. Moreover, some authors did not find evidence of a significant correlation between the two variables (Aivazian et al. 2003; Adil et al. 2011).

### 3.1.4. Retained Earnings

Life cycle theory predicts that mature companies tend to pay more dividends. In fact, DeAngelo et al. (2006) indicate that the ratio of retained earnings to total equity is a good measure to assess the stage of development of firms and to analyze the degree of use of internal or external funds. Mature companies have high levels of retained earnings and their ability to generate cash exceeds their growth opportunities, which decline over time. Thus, their ability to pay dividends is more important compared to companies in development stages. Their results showed that companies pay more dividends when the level of retained earnings is high compared to the amount of equity. Another study, conducted by Denis and Osobov (2008), corroborated these results by using the same ratio in six different financial markets. The study of Al-Sawalqa (2021) also confirmed these results.

### 3.1.5. Firm Age

Von Eije and Megginson (2008) suggested that firm age is a better proxy than retained earnings for assessing firms' maturity. Some studies, such as those of Tamimi and Takhtaei (2014), Mili et al. (2017), Al-Najjar and Kilincarslan (2018) and Eluyela et al. (2019), found a positive relationship between firm age and dividends. These results are mainly

supported by the agency costs and life cycle theories. Nonetheless, other researchers found a negative correlation, such as Al-Sabah (2015). This result is explained by signaling theory based on information asymmetry problems. Moreover, other authors did not find a significant correlation between firm age and dividends (Lumapow and Tumiwa 2017; Al-Sawalqa 2021).

In order to test this variable, we used the following proxy: firm age calculated based on the date of its creation (Ali et al. 2017; Mili et al. 2017; Sari 2018; Zaitul and Pratiwi 2019).

### 3.1.6. Leverage

Various studies supported the negative relationship between leverage and dividends, including those of Bradley et al. (1998), Aivazian et al. (2003), Kumar (2003), Kania and Bacon (2005), Al-Malkawi (2007), Al-Kuwari (2009) and Al-Najjar and Kilincarslan (2018).

Conversely, other empirical studies, such as that of Rehman (2012), found a positive relationship. However, other researchers, including Ho (2003), Abor and Bokpin (2010) and Mehta (2012), noted the absence of a correlation between the two variables.

We chose two ratios to test leverage, as used by Frank and Goyal (2003): total debt/(total debt + book value of equity); and total debt/(total debt + market capitalization). The use of two ratios will be beneficial to test the robustness of our results, because we found it to be insignificant when using the first proxy of dividend distribution in our study.

### 3.1.7. Growth Opportunities

Life cycle theory (DeAngelo et al. 2006) predicts that mature companies present limited growth opportunities but generate high profits and can, therefore, pay more dividends. Fama and French (2001), Kania and Bacon (2005), Amidu and Abor (2006), Abor and Bokpin (2010), Coulton and Ruddock (2011), Rehman (2012) and Danila et al. (2020) all found a negative relationship between dividend payments and growth opportunities. In contrast, Aivazian et al. (2003) and Grace et al. (2019) showed a positive correlation between the two variables, while studies conducted by D'Souza and Saxena (1999), Bisschop (2014) and Tahir and Mushtaq (2016) did not find a significant relationship.

We used the market to book ratio (M/B) as a proxy of growth opportunities (D'Souza and Saxena 1999; Fama and French 2001; Aivazian et al. 2003; Amidu and Abor 2006; Rehman 2012; Danila et al. 2020).

### 3.1.8. Price to Earnings

The results of Myers and Bacon (2004) indicate that companies with a high P/E ratio are more likely to increase the payments of dividends. Firms with high P/E ratios could be considered less risky and would probably have easier access to external funds. The study by Krishnan and Chen (2017) also found a positive correlation between the two variables. However, Franklin and Muthusamy (2010) showed a negative correlation between the payout ratio and P/E.

### 3.1.9. Dummy Variable for Financial Firms

Since we included 18 financial companies (banks, insurance companies and credit institutions) in our sample, we introduced a dummy variable to control the effect of the financial industry on dividend policy. The variable takes a value of 1 when it is a financial company and a value of 0 otherwise. It should also be noted that financial companies have specific regulations in terms of equity management, which may impact their dividend policy.

### 3.2. Sample and Regression Model

The selected sample comprised 70 companies listed on the Casablanca stock exchange during the period from 2003 to 2018. The choice of this long period was motivated, on one hand, by the small number of companies listed on the Casablanca stock exchange and, on the other hand, by the aim to increase the number of observations in order to obtain robust

statistical tests. Furthermore, the choice of a long period makes it possible to reduce the impact of economic growth cycles, which could affect dividend policy globally.

The following Table 1 presents the sample and the databases used.

**Table 1.** Sample and databases.

| | | Databases |
|---|---|---|
| Number of firms and observations | 70 firms<br>859 observations for the dividend yield proxy<br>854 observations for the payout ratio proxy | Bloomberg, annual reports, financial statements, official company websites, information notices and the official website of the Casablanca Stock Exchange: casablanca-bourse.com |
| Period | From 2003 to 2018 (16 years) | |

Source: Authors.

To test our panel data, we adopted the following regression model:

**Y = a + B * (firm age, retained earnings, profitability, leverage, growth opportunities, P/E, firm size, dummy variable for financial companies) + ε**

a: the constant;
B: the coefficient corresponding to each independent variable.
This model tested panel data using linear regressions in order to determine the significant independent variables.

*3.3. Expected Correlations between the Independent Variables and Dividend Distribution*

On the basis of the various theories presented above, we detail, in the following Table 2, the various hypotheses of the expected correlations between each independent variable and dividend distribution.

**Table 2.** Hypotheses of the predicted correlations between the independent variables and dividend distribution.

| Theory | Significant Variables Predicted by Each Theory | Expected Correlation between the Dependent Variable and the Independent One |
|---|---|---|
| Signaling theory | Growth opportunities<br>Profitability<br>Size<br>Age | Positive<br>Negative<br>Negative<br>Negative |
| Agency theory | Growth opportunities<br>Profitability<br>Size<br>Age | Negative<br>Positive<br>Positive<br>Positive |
| Transaction costs and financial flexibility theory | Growth opportunities<br>Leverage<br>P/E | Negative<br>Negative<br>Positive |
| Life cycle theory | Growth opportunities<br>Profitability<br>Size<br>Age<br>Retained earnings | Negative<br>Positive<br>Positive<br>Positive<br>Positive |

Source: Authors.

## 4. Empirical Results

Before proceeding with the regressions, it was necessary to determine the most suitable model for our panel data. We ran the Hausman test in order to choose between the random

effect model and fixed effect model. All Hausman test results indicated that the fixed effect model was the most appropriate one for our data.

### 4.1. Empirical Results Using Dividend Yield

The first regression used the dividend yield proxy and the first proxy of leverage. The results are presented in the following Table 3.

**Table 3.** Empirical results using dividend yield and the first proxy of leverage. *This regression used the fixed effect model. Significant results are followed by one asterisk * if the statistical test is significant with α = 10%; two asterisks ** if the statistical test is significant with α = 5%; and three asterisks *** if the statistical test is significant with α = 1%. The independent variables are firm age (number of years since the creation date of the firm), retained earnings/equity book value, profitability (ROE), leverage using total debt/(total debt + equity book value), growth opportunities (price/book), price to earnings (P/E), firm size using Ln(total assets), dummy variable for financial firms.*

| Dependent Variable: Dividend Yield<br>Independent Variables: | Coefficients (Betas) | t | P > |t| |
|---|---|---|---|
| Firm age | −0.0064828 | −3.84 | 0.000 *** |
| Retained earnings/equity book value | 0.0313535 | 1.84 | 0.067 * |
| ROE | 0.0553176 | 2.24 | 0.026 ** |
| Total debt/(total debt + equity book value) | −0.0407821 | −0.94 | 0.349 |
| P/B | −0.0152298 | −5.17 | 0.000 *** |
| P/E | −0.0002032 | −2.61 | 0.009 *** |
| Firm size (Ln (total assets)) | −0.082662 | −4.15 | 0.000 *** |
| Dummy for financial firms | Omitted due to collinearity | | |
| constant | 2.232905 | 6.01 | 0.000 *** |
| Coefficient of determination $R^2$ (within) | 0.1678 | F-Test | 22.53 |
| | | Probability > F | 0.000 *** |
| Number of observations | 859 | | |
| Number of firms | 70 | | |

Source: Stata software.

To corroborate the notion that leverage is not significant, we used a second proxy for this variable. Thus, Table 4 that follows presents the results of the second regression using dividend yield and the second proxy of leverage.

**Table 4.** Empirical results using dividend yield and the second proxy of leverage. *This regression used the fixed effect model. Significant results are followed by one asterisk * if the statistical test is significant with α = 10%; two asterisks ** if the statistical test is significant with α = 5%; and three asterisks *** if the statistical test is significant with α = 1%. The independent variables are firm age (number of years since the creation date of the firm), retained earnings/equity book value, profitability (ROE), leverage using total debt/(total debt + market capitalization), growth opportunities (price/book), price to earnings (P/E), firm size using Ln (total assets), dummy variable for financial firms.*

| Dependent Variable: Dividend Yield<br>Independent variables: | Coefficients (Betas) | t | P > |t| |
|---|---|---|---|
| Firm age | −0.0064957 | −3.86 | 0.000 *** |
| Retained earnings/equity book value | 0.0294169 | 1.72 | 0.085 * |
| ROE | 0.0592628 | 2.41 | 0.016 ** |

**Table 4.** *Cont*.

| Dependent Variable: Dividend Yield | Coefficients (Betas) | t | P > |t| |
|---|---|---|---|
| **Independent variables:** | | | |
| Total debt/(total debt + market capitalization) | 0.0753526 | 1.57 | 0.118 |
| P/B | −0.0157983 | −5.59 | 0.000 *** |
| P/E | −0.0002141 | −2.76 | 0.006 *** |
| Firm size (Ln (total assets)) | −0.093301 | −4.80 | 0.000 *** |
| Dummy for financial firms | Omitted due to collinearity | | |
| constant | 2.431025 | 6.68 | 0.000 *** |
| Coefficient of determination R² (within) | 0.1695 | F-Test | 22.80 |
| | | Probability > F | 0.0000 *** |
| Number of observations | 859 | | |
| Number of firms | 70 | | |

Source: Stata software.

After running our regressions on the first proxy of dividend policy, we opted for a second proxy in order to confirm or refute our results.

*4.2. Robustness Tests Using the Second Proxy of Dividend Distribution: Payout Ratio*

To test the robustness of our statistical results, we used a second proxy of dividend distribution, which was the payout ratio. The following Table 5 presents the results of all our regressions using the two proxies of dividend distribution.

**Table 5.** Synthesis of the empirical results. These regressions used the fixed effect model. Significant results are followed by one asterisk * if the statistical test is significant with α = 10%; two asterisks ** if the statistical test is significant with α = 5%; and three asterisks *** if the statistical test is significant with α = 1%.

| Independent Variables | Dependent Variable: Dividend Yield | | Dependent Variable: Payout Ratio | |
|---|---|---|---|---|
| | **First Proxy of Leverage (a)** | **Second Proxy of Leverage (b)** | **First Proxy of Leverage (a)** | **Second Proxy of Leverage (b)** |
| Firm age | **Negative *** | **Negative *** | **Negative *** | **Negative *** |
| Retained earnings | Positive * | Positive * | Positive | Positive |
| ROE | Positive ** | Positive ** | Negative | Negative |
| Leverage | Negative | Positive | Negative | Positive |
| P/B | **Negative *** | **Negative *** | **Negative *** | **Negative *** |
| P/E | Negative *** | Negative *** | Positive | Negative |
| Firm size | **Negative *** | **Negative *** | **Negative *** | **Negative *** |
| Dummy variable for financial firms | Omitted due to collinearity | | | |
| R² (within) | 16.78% | 16.95% | 13.89% | 14.12% |
| Number of observations | 859 | 859 | 854 | 854 |
| Number of firms | 70 | 70 | 70 | 70 |
| Fisher test | Significant *** | | | |

Source: Stata software. (a) First proxy of leverage = total debt/(total debt + equity book value); (b) Second proxy of leverage = total debt/(total debt + market capitalization).

## 5. Discussion

### 5.1. Discussion of the Dividend Yield Regressions

First, we note that the Fisher test is significant. This result confirms that at least one of the independent variables has a significant statistical correlation with the dependent variable and that all individual results of the independent variables are valid statistically.

First, we found that firm age presents a significant negative correlation with the dependent variable (Al-Sabah 2015). Signaling theory predicted this correlation and suggested that older companies, which are more mature, suffer less from information asymmetry problems (Fama and French 2001) due to the interest granted to them in the stock exchange, the high number of analysts who follow them, the more effective means of communication available to them and the large number of investors who may be interested in buying their shares. Thus, these firms do not need to use dividends as a signal to be sent to the financial market. Nonetheless, we should note that on the Casablanca stock exchange, the average firm age was 50 years old and the median was 46 years old. In addition to this, among the 859 observations, only nine observations showed an age of less than 10 years old. This situation indicates that the majority of firms could have already reached an advanced stage of maturity.

The significant positive correlation between retained earnings and dividend yield is explained by life cycle theory, which predicts that mature companies, which have accumulated significant retained earnings, have low growth opportunities and pay more dividends (DeAngelo et al. 2006; Denis and Osobov 2008; Al-Sawalqa 2021).

The third significant variable is profitability, which presented a significant positive correlation with dividend payments. This result has been found by several previous studies, such as those of Aivazian et al. (2003), Eluyela et al. (2019), Grace et al. (2019) and Danila et al. (2020). This correlation was predicted by agency theory and life cycle theory. In this regard, agency theory (Jensen and Meckling 1976; Jensen 1986) suggests that more profitable companies, which generate more internal cash flow, would have to pay more dividends in order to reduce agency costs. Moreover, life cycle theory (Grullon et al. 2003; DeAngelo et al. 2006) predicts that more profitable companies are more mature, show fewer growth opportunities and pay more dividends.

The growth opportunities variable also presented a significant negative correlation with dividends, which is supported by different previous researchers, including Fama and French (2001), Kania and Bacon (2005), Amidu and Abor (2006) and Danila et al. (2020). This result was predicted by four theories. First, agency cost theory (Jensen and Meckling 1976; Jensen 1986) assumes that mature firms, which have greater agency cost constraints due to their large and complex structures, pay higher dividends in order to reduce internal cash flow. The financial flexibility theory suggests also that firms with higher growth opportunities would prefer to use internal cash flow to finance their development and reduce dividend payments (Holder et al. 1998). As for life cycle theory, it predicts that mature companies have low levels of growth opportunities and can pay more dividends (Grullon et al. 2003; DeAngelo et al. 2006).

The fifth significant variable is P/E, which indicates that companies with higher P/E pay low amounts of dividends. It should be specified that the P/E ratio provides information about risk and that a higher P/E would indicate that the firm is less risky. Therefore, riskier firms with low levels of P/E would pay fewer dividends, since they would have to pay higher costs for external funding. However, the statistical correlation found was significantly negative. One of the possible interpretations might converge with the result obtained from the relationship with growth opportunities. Companies with a high P/E may be unprofitable or may present high growth opportunities and, therefore, would pay low amounts of dividends. For example, if a company shows a very limited annual profit, its P/E can easily be overestimated or overvalued and reach very high values. Indeed, the average P/E ratio for some firms in our sample reached 26, which means that the investor would need 26 years of profits to recover an initial investment.

Size presented a significant negative result, which has been found by some previous studies, such as those of Javid and Ahmed (2009), Al-Shubiri et al. (2012) and Nazir et al. (2012). The theory that supports this result is signaling theory (Bhattacharya 1979; John and Williams 1985), which indicates that larger companies are generally mature and suffer less from information asymmetry problems. These companies have, then, a lesser need to send a signal to the market via dividend payments.

Leverage did not show a significant result and the dummy variable used for financial companies was omitted from the model, due to collinearity problems. This means that the other independent variables already included the impact of this variable on dividend policy.

In contrast to the two previous studies conducted on the Moroccan stock market, that of Aguenaou et al. (2013) used size, leverage and earnings per share as control variables and their study was limited to a period of 7 years (441 observations). They found a coefficient of correlation (within) limited to 7.63% (compared to 17% in our model) and none of the control variables was significant. Moreover, the research of Mossadak et al. (2016) was limited to a 3-year period (143 observations) and their results indicated that leverage has a significant negative correlation with dividend payments and the two other firm characteristic variables (leverage and size) do not have a significant relationship with dividend policy. However, their study used the ordinary least squares (OLS) model, which is less suitable for panel data. In addition to this, the two studies used only one proxy for dividend distribution. In fact, our research studied firm characteristics' effect on dividend policy using a larger period of investigation (16 years), a larger number of observations, a fixed effect model based on the Hausman test and two proxies of dividend distribution. Therefore, our results could be more robust statistically and provide a better understanding of dividend policy on the Casablanca stock exchange.

It should also be noted that these results are very specific to the Moroccan stock market. Indeed, studies that show a negative correlation between dividends and firm age are rare. In fact, we found only one study that revealed this type of correlation. Nevertheless, this result could be corroborated by the use of the proxy of firm size (which also gives information about firm maturity), which also showed a negative relationship with dividend distribution.

On the other hand, the use of testing performed over a large period of time in our sample has the advantage of limiting the potential impact of economic cycles on dividend policy on the Casablanca stock exchange. Indeed, Morocco experienced known expansion and recession cycles in the period 2003–2018. From 2003 to 2008, Morocco experienced an expansion period before the financial crisis, and 4 to 5 years following the impact of this crisis, the economy started to recover slowly until 2018. Therefore, this long period allowed us to reduce the effect of these cycles on dividend policy in the financial stock market.

The coefficient of determination $R^2$ was equal to 16.78%. This result means that almost 17% of dividend payments were explained by our model using the proxy of dividend yield.

In order to corroborate the insignificant correlation between leverage and dividend policy, we used a second proxy of leverage, which is as follows: total debt/(total debt + market capitalization). The analysis of the results presented in Table 4 above indicates that even when introducing a second proxy for leverage, this variable remained insignificant, as observed in the results of the first regression (Table 3). Furthermore, all the significant variables (firm age, retained earnings, profitability, growth opportunities, P/B, P/E, firm size) remained so in this second regression, with the same type of correlation.

### 5.2. Potential Endogeneity Problems

Dividend policy is a complex financial decision that could present endogeneity problems. Indeed, dividend policy could impact one or more of the independent variables, such as growth opportunities. Firms with higher dividend payments could influence the growth opportunities ratio because, if firms pay higher dividends, it would suggest to the financial market that they do not present sufficient profitable opportunities.

Moreover, as presented in the literature, other variables not used in our research could have a significant effect on dividend policy, such as the ownership structure, board of

directors and CEO characteristics. In our results, the constant presents a significant result, which could indicate that other variables might offer further explanation of the dependent variables' variations. Indeed, previous researchers conducted empirical studies to test these variables (Al-Ghazali 2014; McGuinness et al. 2015; Mili et al. 2017; Bataineh 2020). Therefore, the use of other variables in future studies in the Moroccan stock market might improve our overall understanding of dividend policy in the Moroccan stock market.

Nevertheless, in our study, we used the fixed effect model, which reduces endogeneity problems (Li 2016). In addition to this, we tested eight different independent variables that would potentially limit endogeneity problems, because some variables could have an influence on dividend policy and other independent variables, such as firm size or leverage. Indeed, firm size could influence dividend policy, but it could also have an impact on other variables, including profitability or growth opportunities. Moreover, leverage could influence other variables, such as profitability or P/E. Indeed, the introduction of different time-variant variables, such as leverage or growth opportunities, would help to reduce endogeneity problems.

*5.3. Robustness Tests Using Second Proxy for Dividend Distribution*

The use of the payout ratio enabled us to test the statistical robustness of our results in the first two regressions. Thus, the variables of retained earnings, profitability and P/E showed significant results in the first two regressions using the dividend yield proxy, yet the regressions performed on the payout ratio did not confirm these results. Indeed, profitability did not show a significant result, which could be explained by the potential link between the market value of the share price used in the denominator of the dividend yield ratio and the book value of the share the denominator of the ROE ratio, whereas the payout ratio proxy does not include the value of the share in its calculation. In addition to this, the P/E variable did not display a significant correlation. In this regard, the correlation between P/E and dividend yield may be partly due to the use of the market value of the stock in the denominator of the dividend yield ratio and in the numerator of the P/E. The payout ratio proxy, which is not based on the market value of the stock, did not show any correlation with the P/E variable. Moreover, the retained earnings variable no longer presented a significant result. As such, it should be specified that this variable was significant with only a 90% confidence interval in the previous regressions. Moreover, the result of leverage confirmed the absence of a significant correlation between this variable and the two proxies of dividends. Therefore, these three variables did not show sufficient statistical evidence to be considered as significant variables of dividend policy on the Casablanca stock exchange.

Furthermore, only three variables, among the six significant variables obtained from the first two regressions, remained significant when using the payout ratio proxy for dividend payments. These three variables were firm age, growth opportunities and firm size. They also showed the same type of correlation with the second proxy of dividend distribution, which corroborated our initial findings and the explanations provided for each significant correlation, as described in Section 5.1 above.

Moreover, the $R^2$ showed a slight decrease, from 0.17 in the first two regressions that used the dividend yield proxy to 0.14 in those using the payout ratio. This situation indicates that our model better explained the variations in the first proxy of dividend distribution. In addition to this, the Fisher test remained significant in all regressions, indicating that at least one of the independent variables was significant.

Nevertheless, each stage of research could be improved and enriched through subsequent studies. Therefore, to further investigate dividend policy on the Casablanca stock exchange, future research could include new variables, such as the board of directors' structure (Al-Najjar and Hussainey 2009; Litai et al. 2011), ownership structure (Allen et al. 2000; Manos 2003; Goergen et al. 2005; Le and Le 2017; Jiraporn et al. 2019) and CEO characteristics (Al-Ghazali 2014; McGuinness et al. 2015), which could present new explanations for dividend policy on the Casablanca stock exchange.

## 6. Conclusions

In this article, we discussed several dividend policy theories, which are the signaling theory, agency theory, transaction costs and financial flexibility theory and life cycle theory. Each of these theories aims to explain why firms pay or do not pay dividends.

Then, we presented different empirical results of previous studies, before detailing our sample, the variables' proxies and the predicted correlations between the independent variables and dividend policy based on each theory. Afterwards, we conducted four regressions using two different proxies for dividend payments to test the robustness of our empirical results: dividend yield and payout ratio.

During the period from 2003 to 2018 on the Casablanca stock exchange, we found that the three following variables were significant: firm age, growth opportunities and firm size. They all displayed a negative correlation with the dependent variable of dividend distribution.

The theory that predicted a negative correlation between firm age and size with the dependent variable of dividends was signaling theory. Indeed, older and larger companies suffer less from information asymmetry problems, and they therefore have a lesser need to send a signal to the financial market through dividends. However, this theory did not prevail, given the existence of another significant variable, which was growth opportunities, which showed a negative correlation with dividend payments. As such, signaling theory did not explain this correlation, and the theories that predict a negative correlation between the two variables were agency theory, the financial flexibility theory and life cycle theory. Thus, we can conclude that none of the theories predominated or explained, on its own, dividend policy on the Casablanca stock exchange.

The mobilization of various theories provides a better understanding of dividend payments, which remain a complex financial policy to explain and require further research in the future. However, the introduction of other variables not used in this research, such as the ownership structure, board of directors' structure or CEO characteristics, could offer further insights into dividend policy on the Casablanca stock exchange.

**Author Contributions:** Conceptualization and methodology, R.L. and K.O.; software, R.L.; validation, R.L. and K.O.; writing—original draft preparation, R.L.; writing—review and editing, R.L.; supervision, K.O. All authors have read and agreed to the published version of the manuscript.

**Funding:** This research received no external funding.

**Institutional Review Board Statement:** Not applicable.

**Informed Consent Statement:** Not applicable.

**Data Availability Statement:** Bloomberg database; the official website of the Casablanca stock exchange: casablanca-bourse.com.

**Conflicts of Interest:** The authors declare no conflict of interest.

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
