# Peer review of "Determinants of Dividend Policy: The Case of the Casablanca Stock Exchange"

_jrfm, doi:10.3390/jrfm15120548_

Round 1

Reviewer 1 Report

In terms of contribution, you should first clarify why Moroccan? what's unique and what can be generalized? Since there are no surprising findings, why is Moroccan market provides the research setting to study all these theories? Do these findings push forward our understanding? What should we do with your research? Do you have any suggestions to improve the current regulation or practice? Adding the above discussion and extend your literature review may help you make more contributions and position your contributions better.

The paper seems to claim causality but does not discuss the potential endogeneity issue and its remedies sufficiently. Dividend policy itself is endogenous decision and can affect most of the explanatory variables used in the paper. See Li 2016, Endogeneity in CEO power: A survey and experiment, for a summary of methods to deal with the endogeneity problem. No need to use all these methods but at least discuss them in your scenario. Additionally, the endogeneity problem can be driven by unobservable firm and CEO characteristics you need to discuss. Related to the above point, you should study and rationalize the use of firm size measures in the literature since frim size is the key variable in this area and they affect the independent and dependent variables simultaneously. After all it is the most significant variable in most studies alike. You need to discuss and justify your firm size measure. The results may not be robust to different measures of firm size, which is very common in this area.

Reviewer 2 Report

The reviewed article is, in my opinion, characterised by a satisfactory level of content. Its structure is logical and clear and the individual steps of the study carried out are fully justified. Nevertheless, I would suggest the following corrections and additions:

- based on the numerous literature cited, it would be worthwhile to formulate a research hypothesis or hypotheses,

- the question of the practical and theoretical implications and the highlighting of the novelty and originality of the study carried out (in terms of conclusions, method, etc.) needs to be more fully described and justified, 

- while the study has the advantage of covering a long period (from 2003 to 2018), the macroeconomic situation in Morocco has arguably changed significantly during this time (including external influences), so it would be useful to address whether the strength of the impact of factors in selected sub-periods (defined on the basis of the criterion of the phase of the business cycle or defined as a period of crisis or prosperity) is the same as in the case of a study of the whole period, without distinguishing sub-periods. It would be worth at least commenting on this point, 

- it would be worthwhile to justify the non-inclusion of variables such as e.g. (i) the dominant shareholding of the state, (ii) the free float or the existence of a controlling shareholder, (iii) the share of foreign capital in the shareholding 

- I would suggest adding a description of limitations of the study and directions for further research,

- there are relatively few references to articles from 2020-2022 in the reviewed paper. 

Round 2

Reviewer 1 Report

Well improved.